# Strategies and effectiveness of land decontamination in the region affected by radioactive fallout from the Fukushima nuclear accident: A review

**Olivier Evrard[1], J. Patrick Laceby[2], Atsushi Nakao[3]**

[1]Laboratoire des Sciences du Climat et de l'Environnement (LSCE/IPSL), Unité Mixte de Recherche 8212 (CEA/CNRS/UVSQ), Université Paris-Saclay, Gif-sur-Yvette, France.
[2]Environmental Monitoring and Science Division (EMSD), Alberta Environment and Parks (AEP), Calgary, Alberta, Canada.
[3]Graduate School of Life and Environmental Sciences, Kyoto Prefectural University, Kyoto, Japan.

*Correspondence to*: Olivier Evrard (olivier.evrard@lsce.ipsl.fr).

**Abstract**

The Fukushima Dai-ichi Nuclear Power Plant (FDNPP) accident in March 2011 resulted in the contamination of Japanese landscapes with radioactive fallout. Accordingly, the Japanese authorities decided to conduct extensive remediation activities in the impacted region to allow for the relatively rapid return of the local population. The objective of this review is to provide an overview of the decontamination strategies and their potential effectiveness in Japan, focussing on particle-bound radiocesium. In the Fukushima Prefecture, the decision was taken to decontaminate the fallout-impacted landscapes in November 2011 for the 11 municipalities evacuated after the accident (Special Decontamination Zones – SDZ, 1117 km²) and for the 40 non-evacuated municipalities affected by lower, although still significant, levels of radioactivity (Intensive Contamination Survey Areas, 7836 km²). Decontamination activities predominantly targeted agricultural landscapes and residential areas. No decontamination activities are currently planned for the majority of forested areas, which cover ~75% of the main fallout-impacted region. Research investigating the effectiveness of decontamination activities underlined the need to undertake concerted actions at the catchment scale to avoid the renewed supply of contamination from the catchment headwaters after the completion of remediation activities. Although the impact of decontamination on the radioactive dose rates for the local population remains a subject of debate in the literature and in the local communities, outdoor workers in the SDZ represent a group of the local population that may exceed the long-term dosimetric target of 1mSv $yr^{1}$. Decontamination activities generated ~20 million $m^3$ of soil waste by early 2019. The volume of waste generated by decontamination may be decreased through incineration of combustible material and recycling of the less contaminated soil for civil engineering structures. However, most of this material will have to be stored for ~30 years at interim facilities opened in 2017 in the close vicinity of the FDNPP before being potentially transported to final disposal sites outside of the Fukushima Prefecture. Further research is required to investigate the perennial contribution of radiocesium from forest sources. In addition, the re-cultivation of farmland after decontamination raises additional questions associated with the fertility of remediated soils and the potential transfer of residual radiocesium to the plants. Overall, we believe it is important to synthesize the remediation lessons learnt following the FDNPP nuclear accident, which could be fundamental if a similar catastrophe occurs somewhere on Earth in the future.

**Keywords:** radiocesium; caesium-137; FDNPP; remediation; nuclear accident; Japan.

## 1. Introduction

Large quantities of radiocesium (12–62 PBq) were released into the environment by the Fukushima Dai-ichi Nuclear Power Plant (FDNPP) accident in March 2011 (Stohl et al., 2012;Chino et al., 2011). Airborne and ground contamination surveys demonstrated that the contamination was the highest (i.e., initial $^{137}$Cs levels >100,000 Bq m$^{-2}$) in a plume extending to the northwest of the FDNPP covering an area of ~3000 km² (Kinoshita et al., 2011;Chartin et al., 2013;Yasunari et al., 2011). Although many radioactive substances were released into the environment by the FDNPP accident, radiocesium (i.e., $^{134}$Cs and $^{137}$Cs) presents the most serious risk to the local population over the medium to long term as it was emitted in very large quantities and it has a relatively long half-life (i.e., $^{134}$Cs - 2 years; $^{137}$Cs - 30 years) (Steinhauser et al., 2014).

Numerous investigations have been conducted by Japanese and international researchers to improve our understanding of the fate of radiocesium in the Fukushima region (for a review, see: Evrard et al., 2015). In general, radiocesium sorption mechanisms were characterised (Fan et al., 2014;Nakao et al., 2015) and their fluxes measured in riverine systems draining the main radioactive plume (Nagao et al., 2013). Land use (Koarashi et al., 2012) and soil properties (Nakao et al., 2014) were shown to control the migration of radiocesium in soils. Accordingly, the fate of this contaminant was intensively investigated in forest ecosystems (Gonze and Calmon, 2017) and cultivated landscapes (Yoshimura et al., 2016), which are the two main land uses in the fallout-impacted region. Typhoons and other major rainfall events were also demonstrated to drive soil erosion and sediment migration processes thus directly influencing post-fallout radiocesium dynamics (Chartin et al., 2017).

Between 2011–2018, there were 578 published studies with the keywords 'radiocesium' and 'Fukushima' in the Scopus database (Figure 1). Approximately 90% of these articles were published by Japanese scientists, demonstrating the extensive research effort conducted by the national scientific community in Japan on the processes occurring in this post-accidental context. Since the second half of 2013, remediation activities started to be implemented under the supervision of the Japanese authorities to decontaminate soils. These activities have significantly affected the spatial and temporal redistribution of radionuclides in the Fukushima-impacted area. As decontamination is now completed in many regions and more than 50 scientific studies have been conducted on different aspects of these operations (Figure 1), synthesizing the results obtained by this applied research is important for the scientific community. Of note, this review will not synthesize non-peer reviewed reports published by the Japanese authorities, although numerous resources are available on the official websites of multiple Japanese ministries (Table 1).

Although radiocesium is mainly transported in particle-bound form in the Fukushima fallout-impact area (Konoplev et al., 2016), dissolved radiocesium was found in numerous environmental compartments, primarily during the immediate post-accidental phase (Yoshimura et al., 2014). As most of the dissolved radiocesium migrated through these landscapes immediately after the FDNPP accident, this literature review will focus on particulate radiocesium. Furthermore, as $^{134}$Cs and $^{137}$Cs were emitted in equivalent proportions into the environment in March 2011, with an initial $^{134}$Cs/$^{137}$Cs activity ratio of ~1 (Kobayashi et al., 2017), this review will focus primarily on $^{137}$Cs owing to its longer half-life and thus greater risk to the local population over the medium to long-term. Accordingly, the goal of this review is to examine the remediation strategies and their effectiveness for particulate bound $^{137}$Cs in Japan.

This literature review will be divided into five main sections. First, the spatial extent of the decontaminated zone and the schedule of these remediation activities will be outlined. Second, the remediation strategies in different environments (i.e., farmland, river, forests) will be presented along

with a summary of their cost effectiveness. Third, the impacts of remediation activities on dosimetry will be summarized. Fourth, the initiatives to manage the large volume of waste generated by remediation will be discussed. Fifth, major research questions and requirements to guide the future management of Fukushima fallout-impacted areas will be identified and presented. The objective of this review is to provide a synthesis of the remediation lessons learnt in Japan following the FDNPP nuclear accident, which are fundamental in light of the potential for a similar catastrophe to occur somewhere on Earth in the future (Christoudias et al., 2014).

## 2. Areas targeted by decontamination

In November 2011, the Japanese government adopted the *Act on Special Measures Concerning the Handling of Pollution by Radioactive Materials* (Japanese Ministry of the Environment, 2011a) in order to reduce the impact of radioactive substances from the FDNPP accident on human health and the environment (Yasutaka and Naito, 2016). In support of this Act, decontamination guidelines were released by the Japanese Ministry of Environment in December 2011 and updated in 2013. These guidelines outlined the methods for surveying and quantifying the levels of contamination and the way to prepare these areas targeted for remediation (Japanese Ministry of the Environment, 2013). A decontamination roadmap (*Policy for Decontamination in the Special Decontamination Area*) was implemented in January 2012 under the direct supervision of the Japanese government.

According to the decontamination roadmap, the remediation programme had to be implemented in 'special areas' where targets were set for the exposure of the public to external dose rates in order for residents to return to their day-to-day lives (Yasutaka and Naito, 2016). Achieving pre-accident radiation levels is not the objective, rather the effectiveness of decontamination will ultimately depend upon the land use and the air dose of each particular area. Two zones were delineated with different strategies for remediation (Figure 2). First, Special Decontamination Zones (SDZ) are areas located within a 20-km radius of the FDNPP or areas where the cumulative dose one year after the accident was expected to exceed 20 mSv $yr^{-1}$. SDZs occur in 11 municipalities (1117 $km^2$) where residents were evacuated after the FDNPP accident in 2011. The central government of Japan is responsible for remediation works in SDZs. Second, Intensive Contamination Survey Areas (ICAs) refer to 102 municipalities from eight Prefectures with ambient dose rates exceeding 0.23 µSv $h^{-1}$, designated as ICAs by the Ministry of Environment on December 28, 2011 (Mori et al., 2017). The area of the ICAs is eight times greater than the SDZ (Yasutaka and Naito, 2016). In particular, the decontamination methods and target areas for remediation in the ICAs differ from those of the SDZ with decontamination activities for the ICAs conducted by local governments with support from the central government. In total, the SDZs and ICAs cover a surface area of 8953 $km^2$ with a population that was not evacuated after the accident of ~1.7 million (Yasutaka and Naito, 2016).

In the literature, there is debate regarding the need to initiate decontamination so quickly after the FDNPP accident (e.g., Yasutaka et al., 2013a). Delaying decontamination could allow for the natural decay of radioisotopes and thus significantly lower the costs of achieving radiation exposure targets. For example, Munro (2013) estimated that the optimal delay for implementing remediation activities was in the range of 3–10 years after the accident, with an optimal delay of 8.8 years.

## 3. Decontamination Strategies and their Cost Effectiveness

The effectiveness of decontamination was assumed to strongly vary depending on the remediation method and the initial radiation dose rates prior to decontamination. Different remediation techniques were proposed depending on the land use and the zone (i.e., SDZ vs. ICA). Yasutaka et al. (2013a) and Yasutaka et al. (2013b) compared the impact of four scenarios of decontamination, including two very unlikely options (i.e., minimal and maximal scenarios), in terms of effectiveness and cost according to the results of demonstration tests conducted by JAEA. These results were updated in a more recent publication (Yasutaka and Naito, 2016). Only the two scenarios following the guidelines provided by the Japanese government are assessed in this review. As such, the results of both the minimal and maximal remediation options are therefore not discussed. In the first scenario, 5 cm of topsoil was removed from 50% of agricultural land in the SDZ where $^{137}$Cs concentrations exceed 5000 Bq kg$^{-1}$, and replaced with a 5-cm layer of 'clean' soil (measure A1; Table 2). In the remaining 50% of agricultural land with $^{137}$Cs concentrations below 5000 Bq kg$^{-1}$, topsoil was interchanged with subsoil (measure A3). Ploughing with zeolite and potassium (measure A4) was adopted for agricultural land in ICAs where the annual additional effective dose exceeded 1 mSv yr$^{-1}$. In the second scenario, measure A1 was applied to all cultivated land. Similar measures (e.g. Table 2) were included in both scenarios to decontaminate forested areas, roads and houses. The total decontamination cost for implementing these remediation measures varied between 2–5.1 trillion JPY (~16–41 billion EUR), with 1.3–2 trillion JPY (~10–16 billion EUR) for the SDZ and 0.7–3.1 (~6–25 billion EUR) for the ICAs. Although the area where decontamination has been implemented in the ICAs (922–3330 km²) covers a surface 3 to 11 times larger than that of SDZ (295 km²), the decontamination costs for the SDZ and ICAs are in the same order of magnitude.

The decontamination program includes a variety of other activities on top of the actual on-site remediation works, including the transport of waste, the volume reduction of waste along with the temporary, the interim, and the final storage of decontamination waste and removed soil in containers. Depending on the set of measures implemented in the field, the cost of the remediation works will therefore be highly variable. A synthesis of the unit costs for waste management and storage is provided in Table 3 (Yasutaka and Naito, 2016). As shown by Yasutaka et al. (2013b), the quantity of waste generated when decontaminating agricultural land varies considerably depending on the decontamination method used. Consequently, differences in the quantity of waste generated resulted in large differences between agricultural land decontamination methods and the costs associated with storage containers, temporary storage sites and interim storage facilities.

According to the latest available figures from the Japanese Ministry of the Environment (2019b) at the end of 2018, the volume of soil waste generated in the SDZ was 9,100,000 m$^3$ with a remediation cost of approximatively 1.5 trillion JPY (~12 billion EUR). In the ICAs, the latest figures available for March 2018 showed that 7,900,000 m$^3$ of waste soil were produced with a remediation cost of approximately 1.4 trillion JPY, equivalent to ~11 billion EUR (Japanese Ministry of the Environment, 2019a).

## 4. Strategies for decontamination in various environments, and their effectiveness

In general, owing to the strong and nearly irreversible bond of radiocesium to fine soil particles, the majority of FDNPP derived $^{137}$Cs is stored within the topsoil (i.e. the top 5 cm) in undisturbed soils (Lepage et al., 2014;Matsunaga et al., 2013;Takahashi et al., 2015;Mishra et al., 2015;Matsuda et al., 2015). Mishra et al. (2015) reported that for these undisturbed soils, the vertical migration of radiocesium down the soil profile was slower in forest soils compared to grassland soils. In disturbed soils, anthropogenic activities may increase the depth migration of radiocesium down the soil profile

(Lepage et al., 2015;Matsunaga et al., 2013). For example, Lepage et al. (2015) illustrated that 90% of the FDNPP derived radiocesium was homogeneous throughout the tilled soil layer in cultivated soils. Endo et al. (2013) reported that radiocesium concentrations were not depth dependent in cultivated soils (i.e. paddy fields) whereas they declined exponentially in uncultivated soil. Both Sakai et al. (2014b) and Tanaka et al. (2013) also demonstrated that radiocesium from the FDNPP accident was measureable at 15cm depth in rice paddy fields. As illustrated by Koarashi et al. (2012), the penetration of radiocesium in the soil differed depending on both the land use and the physicochemical properties of the soil (e.g., bulk density, clay content and organic matter content). However, remediation strategies consisting in removing the top 5 cm layer of the soil should have been effective as cultivation or other human activities that may have led to the redistribution of radiocesium with depth after the accident were prohibited in the main fallout impacted region.

*Soil/farmland decontamination*

Different strategies were carried out in Japan to decontaminate soil in farmland, either removing the contaminated layer of soil or through the sowing of plants having the capacity of extracting and concentrating radiocesium from the soil. There are few publications in international journals regarding the potential or effectiveness of the latter strategy (e.g. Pareniuk et al., 2015). Among the few available studies, Kobayashi et al. (2014) grew thirteen plant species from three families (*Asteraceae*, *Fabaceae*, and *Poaceae*) in shallow and deeply cultivated fields where the 0–8 cm and 0–15 cm soil layers were respectively ploughed. The variation in plough depth was expected to reflect the impact of different contact zones between the root systems and radiocesium in the soil. Overall, 29 to 225 Bq kg$^{-1}$ dry weight of $^{137}$Cs were found in the plants, corresponding to transfer factors ranging from 0.019 to 0.13 (geometric mean (GM), 0.057) for plants growing in shallow soils, and from 0.022 to 0.13 (GM 0.063) for plants growing in deeper soils (Kobayashi et al., 2014). The authors found that none of their tested plant species resulted in a significant decrease in radiocesium in soil likely because of the strong fixation of $^{137}$Cs to clay particles. This result was confirmed by Yamashita et al. (2014) who showed that 99 wild plants grown in paddy and upland fields had a very low phytoextraction efficiency. Tamaoki et al. (2016) reached the same conclusions, although they suggested Kochia (*Bassia scoparia*) as a potential candidate for phytoremediation although its efficiency in removing $^{137}$Cs would require numerous cultivation rounds.

Accordingly, given the low efficiency of phytoremediation, the main remediation strategy consists of removing the surface layer of soils with the majority of radiocesium. The effectiveness of this strategy was examined by Sakai et al. (2014a) in Kawamata Town. Approximately 5-10 cm of the surface soil was removed from one rice paddy by heavy machinery, whereas a nearby paddy field was not decontaminated and used as control plot. Both of these paddies then were ploughed and planted with rice. Five surface soil samples (0-5 cm) were collected after decontamination and prior to ploughing on June 12, 2011. Thereafter five soil cores (20 cm depth) were collected on July 13, 2012 at 3 m intervals across both rice paddy fields. In 2011, the accumulation of radiocesium in the 0-5 cm surface layer of the soil in the decontaminated paddy field (170±64 Bq kg$^{-1}$) was lower than the control rice paddy field (2231 ±64 Bq kg$^{-1}$). However, the $^{137}$Cs concentration of the surface soil layer in the decontaminated rice paddy field (753±62 Bq kg$^{-1}$) was significantly higher in 2012 than in 2011. This result suggests that radiocesium is likely redistributed through the rice paddy field irrigation and drainage networks. The authors concluded that the redistribution of soil within the paddy fields may decrease the effectiveness of decontamination. A lack of replicates was outlined by the authors and prevented them to finally conclude on the effectiveness of surface removal for decontamination (Sakai et al., 2014a). In contrast, Kurokawa et al. (2019) observed a 80% decrease in $^{137}$Cs activities after

decontamination in cultivated land of Tomioka Town, showing the efficiency of this remediation strategy.

Another study was conducted in experimental paddy fields located ~40 km from the FDNPP (Wakahara et al., 2014). Two plots were established: a paddy field where the top 5–10 cm of soil was removed before cultivation and a control paddy. The $^{137}$Cs soil inventory measured 3 months after the FDNPP accident was approximately 200,000 Bq m$^{-2}$. However, after decontamination, this inventory decreased to 5000 Bq m$^{-2}$. Suspended sediment and $^{137}$Cs fluxes were measured in the outflow of the paddy fields after puddling (i.e., the mixing of soil and water before planting rice) and they were 11.0 kg and 630,000 Bq (1240 Bq m$^{-2}$) respectively in the control paddy, versus 3.1 kg and 24,800 Bq (47.8 Bq m$^{-2}$) in the decontaminated paddy. After irrigation, 5.5 kg of particles and 51,900 Bq (102 Bq m$^{-2}$) of $^{137}$Cs were discharged from the control plot, whereas 70 kg of suspended sediment and 165,000 Bq (317 Bq m$^{-2}$) of $^{137}$Cs were discharged from the remediated field. This 3-fold higher export of $^{137}$Cs from the decontaminated paddy was likely explained by the supply of contamination from upper paddy fields, which remained connected to the remediated field. This result highlights the importance of remediation strategies focusing on the entire catchment scale.

Although this practice has not been specifically investigated in the literature, decontamination of farmland in the Fukushima fallout-impacted region was not limited to the removal of the 5-cm topsoil layer concentrating the radiocesium. After this first step, a layer of crushed granite, extracted from local quarries dedicated to decontamination, is used to replace the removed soil layer (Evrard et al., 2019). The entire soil profile consisting of the residual initial soil (in depth) and this crushed granite layer (on top) is then thoroughly mixed to prepare the soil for recultivation with the objective being to further dilute the residual radiocesium activities in the soil (Figure 3).

Other studies investigated the impact of remediation works on the radiocesium levels measured in sediment transiting the river networks draining the main radioactive pollution plume. Evrard et al. (2016) modelled the progressive dilution of radiocesium concentrations measured in sediment following decontamination works. They demonstrated a 90% decrease of the contribution of upstream contaminated soils to sediment transiting the coastal plains of the Mano and Niida Rivers between 2012 and 2015. Furthermore, Osawa et al. (2018) monitored the radiocesium concentrations in suspended sediment collected in two tributaries of the Mano and Niida Rivers from 2013 and 2016. They also attributed a decrease in the $^{137}$Cs concentrations observed in 2016 to the decontamination efforts completed in 2015 in the local catchments.

*River channel decontamination*

Riverside parks and playgrounds are popular across Japan. Sediment containing high quantities of radionuclides may also accumulate near these parks and playgrounds in river channels and floodplains following flooding events (Saegusa et al., 2016). However, sediment deposition is highly heterogeneous both horizontally and vertically across floodplains. Furthermore, sediment deposited in the river channel may be resuspended during subsequent flood events. In these conditions, the common decontamination guidelines (i.e. removing the uppermost layer; Table 2) are difficult to implement effectively.

Nishikiori and Suzuki (2017) investigated this challenge in the 13-km² Kami-Oguni River catchment, a tributary of the Abukuma River, in the Fukushima Prefecture. Decontamination of a 170-m long and 8 to 13-m wide river section isolated from the floodplain with 2-m high concrete dikes was studied, as the roads located on top of the banks were used by children to go to school. First, all the plants were

removed from the channel. Then, the top 5-cm layer of sediment was excavated from the dike slopes and planted with turf. Afterwards, sediment was removed from the channel, and the removal depth (between 15-35 cm) was locally adjusted depending on the vertical distribution of radiocesium measured using a NaI scintillation detector before, immediately after, and then three months after the remediation campaign. In addition, sediment samples were collected along transects at various depths in the floodplain and analysed with coaxial germanium detectors. Radiocesium contamination strongly varied with depth, depending on changes in the mud versus sand fractions, the former being enriched in radiocesium. Radiocesium concentration also varied across the channel, depending on the local flow velocity, which varied depending on the flood magnitude, the plant density and the microtopography. Before decontamination, air dose rates 1 cm above the ground varied between 0.2–1.99 µSv $h^{-1}$, demonstrating the heterogeneity of contamination. After remediation, the contamination levels decreased by a factor of approximately 2, from a mean of 0.78 (±0.41) µSv $h^{-1}$ before decontamination to 0.34 (±0.15) µSv $h^{-1}$ after decontamination at 1 cm above the ground. However, Nishikiori and Suzuki (2017) underlined the risk associated with the potential deposition of contaminated material originating from upstream landscapes during subsequent flood events.

*Forest decontamination*

The guidelines for the decontamination of forested areas in the Fukushima Prefecture (Table 2) indicate that only those areas lying within 20 m of houses should be targeted for remediation (Yasutaka and Naito, 2016) (Figure 4). Although the remediation in forests has not been a priority for the Japanese authorities during the early post-accidental phase, pilot studies were conducted to quantify the potential effectiveness of wider remediation programmes. Ayabe et al. (2017) investigated the impact of local-scale decontamination including the removal of the litter layer, the superficial soil layer, and the understorey in a secondary mixed forest with a cover of bamboo grass, *Sasa nipponica,* as understorey located in Kawamata Town. Although the total $^{137}$Cs contamination in soil and litter was reduced by ~20% after decontamination compared to an adjacent untreated area, the radioactive contamination levels returned to their initial level four months after the completion of remediation works. This was likely due to the occurrence of a torrential rainfall event and the supply of contaminated foliage to the ground by litterfall. These results suggest that the removal of the litter and superficial soil layers in a contaminated forest may have limited effectiveness.

In another study by Lopez-Vicente et al. (2018), several different forest decontamination practices were compared through the monitoring of $^{137}$Cs contamination in soil and leaf samples in 10 plots installed in the evacuation zone, 16 km to the southwest of FDNPP, between May 2013 and July 2015 (i.e. 27 months monitoring). Four potential forest remediation strategies were assessed. First, the combination of tree thinning and litter removal provided the best results to reduce $^{137}$Cs export (350–380 Bq $m^{-2}$ $day^{-1}$), followed by the application of tree thinning only (163–174 Bq $m^{-2}$ $day^{-1}$). Clearcutting and litter removal provided limited results (92–104 Bq $m^{-2}$ $day^{-1}$), with higher $^{137}$Cs export rates than those observed from the control plots (52 Bq $m^{-2}$ $day^{-1}$). Finally, plots where tree matting was conducted had lower $^{137}$Cs export rates (19–25 Bq $m^{-2}$ $day^{-1}$) than control plots. Overall, the decreasing trend in radiocesium concentrations measured in the plot outflow was high in 2013, moderate in 2014 and low in 2015 owing to the vegetation recovery after the countermeasures.

## 5. Impact of decontamination on dosimetry

Two parameters are assessed before authorizing evacuees to return home: the prevalent dose rate and the cumulative dose. Importantly, background radiation levels need to be incorporated into this assessment. In the Fukushima Prefecture, background does rates before the FDNPP accident were

estimated to be ~0.04-0.05 mSv h$^{-1}$ (National Institute of Advanced Industrial Science and Technology, 2011).

Individual external radiation doses (mSv day$^{-1}$) may not be directly related to outdoor air doses (mSv day$^{-1}$) as people do not spend 24 hours a day outside. When people are inside, the distances from the sources of radiation are greater and walls generate a shielding effect (IAEA, 2000). In Japan, when the Ministry of the Environment estimated daily individual external effective dose rates, it was assumed that people spent 8 hours outdoors and 16 hours indoors, with the indoor air dose rate being 40% of the outdoor air dose rate (Japanese Ministry of the Environment, 2013). Based on these assumptions, the external radiation dose rate is 60% of the air dose rate. Several researches have estimated external conversion coefficients according based on data provided by the Ministry of the Environment (Yasutaka and Naito, 2016).

Figure 5 compares the annual and individual dose rates that the global population may be exposed to in order to help facilitate a comparison with levels in the FDNPP fallout-impacted region. In Japan, a long-term dosimetric target of 1mSv yr$^{-1}$ was adopted by the Nuclear Emergency Response Headquarters. Accordingly, a guidance value of 0.23 µSv h$^{-1}$ was proposed to achieve the target by implementing decontamination measures. In particular, areas with ambient dose rates exceeding this value were defined as ICAs. This guidance value is based on a simplified deterministic model assuming that inhabitants again spend 8 hours outdoors and 16 hours indoors (i.e. a shielding factor of 0.4) per day and that the contribution of natural radiation is 0.04 mSv h$^{-1}$ (IAEA, 2013). According to Mori et al. (2017), this model has three main challenges. First, the same behavioural pattern is assumed for the entire population. Second, the radiation exposure is assumed to be uniform. Third, conservative assumptions are adopted when converting the ambient dose into an effective dose. For instance, the time spent outside is assumed to be 8 hours, which is more than anticipated for the majority of the population and likely results in an overestimation of the actual measured doses (Nomura et al., 2015).

Although this approach is effective for the immediate post-accidental context, more sophisticated approaches are required to estimate doses over the longer term. Therefore, a probabilistic method that accounts for spatial variations (i.e., houses, workplaces, and other environments) of the contamination and for inter-populational variations (i.e., indoor workers, outdoor workers, pensioners) in behavioural patterns was developed by Mori et al. (2017). For this approach, the 95$^{th}$ percentile doses for outdoor workers were above 1 mSv yr$^{-1}$ in 25 of the 59 municipalities in Fukushima Prefecture (1–35 mSv yr$^{-1}$). In particular, the doses to more than 90% of the outdoor workers in Okuma town, Futaba town, Tomioka town, Namie town, and Iitate village were over 1 mSv yr$^{-1}$. Furthermore, the 95$^{th}$ percentile doses for indoor workers were above 1 mSv yr$^{-1}$ in Okuma town, Futaba town, Tomioka town, Namie town, and Iitate village. If people return home in these municipalities, it is possible that they would be exposed to doses exceeding 1 mSv yr$^{-1}$ for all population groups. However, the results indicate that the same behavioural patterns and contamination levels should not be assumed for all inhabitants nor all municipalities. Based on the different behaviour of the local population, the 95$^{th}$ percentile doses of indoor workers and pensioners in 53 of the 59 municipalities were below the dosimetric target of 1 mSv yr$^{-1}$ (0.026–0.73 mSv yr$^{-1}$) (Mori et al., 2017). Radiation dose rates were also measured among different types of workers having professional activities in the village of Kawauchi where the annual doses of foresters (range: 0.7–1.9 mSv yr$^{-1}$) were not significantly higher than those of farmers (0.7–1.5), builders (0.6–1.5), office workers (0.5–1.5) and unemployed individuals (0.5–1.7). In contrast, decontamination workers (0.5–7.1) were found to have significantly higher dose rates (Orita et al., 2017).

The workers involved in decontamination activities were often directly exposed to internal irradiation through inhalation, which is much more difficult to measure than the external irradiation. Accordingly,

eighty-three people who worked in highly contaminated areas where surface $^{137}Cs$ deposition density was over 100 kBq $m^{-2}$ were enrolled in a study (Tsubokura et al., 2013). Using a database on internal exposure from the Hirata Central Hospital in Fukushima Prefecture, data were compiled on age, gender, body weight, equipment used in decontamination activity, total working period, duration between the final working day and the day of an examination, and $^{134}Cs$ and $^{137}Cs$ body burden. Hirata Central Hospital was also equipped with a permanent whole body counter with detection limits of 300 Bq per individual for both $^{134}Cs$ and $^{137}Cs$ measurements following a 2-min scan. The levels of internal radiocesium exposure among all the decontamination workers were below the detection limits. No other radionuclides besides natural $^{40}K$ were detected. No acute health problems had been reported. However, levels of external exposure were not assessed, as individual data on dose rates were not available. This study suggests that the resuspension of radioactive materials may cause a minimal internal contamination during decontamination works (Yamaguchi et al., 2012). Other studies calculated that radiation doses from internal exposure were marginal (Hayano et al., 2013;Tsubokura et al., 2015). As such, remediation efforts should be concentrated on reducing the external exposure of the local population.

According to the decontamination scenarios described in section 3, the reduction in annual individual additional Effective Dose (ED) for all decontamination scenarios was 1666 person-Sv for the SDZ and 876-1245 person-Sv for the ICAs (Yasutaka and Naito, 2016). Despite the higher reduction rate achieved in the SDZ compared to the ICAs, they remained in the same order of magnitude although the decontamination efficiencies were very different in both areas. This result may be directly attributed to the differences in population density in SDZ and ICAs, with 90,000 inhabitants living in SDZs in 2010 versus approximately 1.5 million inhabitants living in ICAs areas exposed to over 1 mSv $yr^{-1}$. This strong dependence of ED on population densities may lead the authorities to concentrate their remediation efforts in the most densely populated areas. The results obtained also depend on the effectiveness of these decontamination programmes. For instance in ICAs, where approximately 1 million inhabitants reside in areas exposed to 1–5 mSv $yr^{-1}$, the reduction in annual individual additional ED was much larger in those areas where the full decontamination scenario would be implemented.

From the aforementioned research on river channel decontamination, the external radiation dose was calculated for paths along the river used by children to go to school and the nearby playgrounds used for outdoor activities incorporating an adapted time of exposition (35 h $yr^{-1}$ for commuting and 24 h $yr^{-1}$ for outdoor activities) (Nishikiori and Suzuki, 2017). After decontamination of the river channel, radiation dose rates decreased by a factor of approximately 2. These authors stated that the optimal strategy should be to reduce the annual individual additional ED as much as possible for the whole population, while also decreasing high dose individuals (Yasutaka and Naito, 2016). Indeed, the authorities should not only assess the cost-benefit effectiveness of remediation programmes, they must also consider ethical and social costs (Oughton et al., 2004).

## 6. Treatment of decontamination waste (soil, vegetation)

*Waste management*

The management of waste generated by the succession of catastrophes that affected the Fukushima Prefecture in March 2011 has proved to be very complex, as debris derived from the earthquake, the tsunami and the radioactive materials were mixed, resulting in a very atypical mixture of 'disaster waste' (Shibata et al., 2012). Earthquake and tsunami-associated waste had elevated levels of metals and metalloids (e.g., mercury, arsenic and lead), with the tsunami waste being particularly difficult to manage.

Regarding waste contaminated with $^{137}Cs$, the final objective is to bring radiocesium to the solution phase and then enrich it, to reduce it to the smallest possible volume. In the Fukushima Prefecture, the radiocesium concentrations found in the disaster waste are lower than other alkali metals. Therefore, the treatment methods require approaches that help concentrate $^{137}Cs$ (Parajuli et al., 2016a). The reduction of solid waste volume can be achieved through compaction or incineration. For organic waste (i.e. forest litter, weeds, wood, or tree branches from contaminated areas), incineration ('thermal treatment') is traditionally preferred (IAEA, 2003, 2006) as it reduces the volume of waste by several orders of magnitude (Parajuli et al., 2013). The problem is that this 'thermal treatment' may enrich contaminants and the Japanese legislation has a 8000 Bq $kg^{-1}$ radionuclide threshold for placing waste in landfills (Japanese Ministry of the Environment, 2011b).

Accordingly, waste contaminated with radionuclide levels between 8000 Bq $kg^{-1}$ and 100,000 Bq $kg^{-1}$ needs to be disposed in designated landfills equipped with radiation level and leachate monitoring as well as a treatment system in order to control the potential release of radioisotopes into the environment (Parajuli et al., 2013). Therefore, either specially designed landfills need to be constructed or pre-treatment methods need to be designed to remove radionuclides from the waste. This issue is crucial as the construction of temporary storage sites and interim storage facilities were estimated to account for 50% of the overall cost of decontamination. For example, transport, storage and administrative costs were estimated to represent a cost of 1.55–2.12 trillion JPY (~12.4–17 billion EUR) for the decontamination scenarios complying with the guidelines of Japanese authorities (Yasutaka and Naito, 2016). Furthermore, securing routes and locations for transporting more than 20 million tonnes of decontamination waste and removed soil that were generated to the interim storage facilities remains a major challenge. Nevertheless, assessing the management and storage of low-concentration radioactive cesium-containing soil and methods for using controlled landfill sites may lead to a significant reduction in the amount of material requiring transport.

The combustible waste generated through decontamination was initially stored at temporary storage facilities (Figure 6). The volume of this waste was to be reduced by incineration, and the incineration ash was transferred to interim storage facilities. In 2013, the Japanese Ministry of Environment made a plan stating that incineration ashes with high $^{137}Cs$ concentrations and leachable characteristics should be stored in concrete shielded structures facilities. After being transferred to interim storage facilities, incombustibles (e.g., soil) were planned to be stored at soil storage facilities in the interim storage facilities (Yasutaka and Naito, 2016). The interim storage facilities are to be built in the areas neighbouring the FDNPP (i.e., in Okuma and Futaba municipalities), while the temporary storage sites were planned to be built in six municipalities in the SDZ (i.e., from North to South: Iitate, Minamisoma, Katsurao, Namie, Tomioka and Naraha municipalities).

Contaminated soil removed by decontamination works is transported to an interim storage facility where flammable decontamination waste (DW) is incinerated or melted to reduce its mass and volume. Depending on its radiocesium content, this waste is either stored at an interim storage facility or disposed in a leachate-controlled type of landfill site (Fujiwara et al., 2017). The total surface area of the interim storage facilities in Futaba and Okuma municipalities is planned to cover ~1600 ha, and by February 2019, a contract was already established between Japanese authorities and landowners for ~70% of the land required for storage. Soil storage operations started in October 2017 in Okuma and in December 2017 in Futaba. By March 2019, ~2.5 million $m^3$ of waste soil had already been transported from the temporary storage facilities distributed across all the remediated area to these two interim storage facilities (Japanese Ministry of the Environment, 2019a). All the soil waste is planned to be transported to the Okuma and Futaba sites by the end of 2021 (Japanese Ministry of the Environment, 2019b). The final disposal of this decontamination waste should take place outside of

the Fukushima Prefecture, within 30 years after the opening of the interim storage facilities (i.e., ~2047).

*Incineration*

The temperature used in the furnaces used for incineration of radioactive waste is similar to that used in the plants treating municipal waste (870–882°C). The incinerators for radioactive waste are radiation-controlled areas, with workers following protocols in accordance with the Ordinance on Prevention of Ionizing Radiation Hazards (Act No. 134 of the 2015 amendment of Law No. 41 of the Japanese Ministry of Labour in 1972). The heavier particles are collected at the bottom of the furnace, generating the so-called bottom ash (BA), while the lighter particles pass to a bag-filter where the temperatures are kept lower (250–300°C) and where the so-called fly ash (FA) and the vaporised cesium are collected (Figure 7). The exhaust gas is filtered to trap the residual fine particles, generating several types of FA. Measurements made on incineration products since 2015 showed that BA and FA are produced with similar levels of radiocesium, both with low radiocesium leachability (<1%) (Fujiwara et al., 2017). Radiocesium levels in the exhaust gases were found to be lower than method detection limits (Parajuli et al., 2013).

*Incineration ash treatment*

The chemical form and the leachability of radiocesium depends on the type of waste incinerated. Results observed for three different types of ash samples suggest that $^{137}Cs$ along with other alkali metals in wood bark and household garbage ashes, originated from burnable materials, were mostly washed out with water even at ambient temperatures. However, municipal sewer sludge was different, with potential $^{137}Cs$ elution only occurring under very specific conditions (i.e., with acid treatment and under high temperatures). Acid treatment at high temperatures was found to be inappropriate for treating wood bark and household garbage ashes because of the generation of a Ca excess leading to gypsum formation and complexifying the subsequent treatment process (Parajuli et al., 2013).

*Soil recycling*

As 22 million $m^3$ of decontamination soil (i.e. 90% of the total) and incineration ash waste (10%) is expected to be produced through remediation of the fallout impacted region, recycling may be instrumental for reducing this volume (Takai et al., 2018). The Japanese Ministry of Environment developed a policy to separate decontamination soil into low- and high-activity soils, the former being ‘recycled’ in public projects. In these uses, decontamination soil will be used for the basic structure and will be covered by uncontaminated soil or concrete. In theory, the unconditional ‘clearance level’ defined by IAEA for the use of recycled material is fixed to 100 Bq $kg^{-1}$ for radiocesium. However, as disaster waste was found with higher $^{137}Cs$ levels, the Japanese Ministry of Environment decided that those materials with radiocesium levels up to 3000 Bq $kg^{-1}$ can be reused at a minimum depth of 30 cm underground (reference level assessed for recycling of concrete for the road subbase course). For decontamination soil recycling, the radioactivity level had to be reanalysed for a different type of engineering structures (deterministic estimation of radiation dose rates). The corresponding level of radiocesium concentrations in the soil was estimated to 6000 Bq $kg^{-1}$. To confine doses to levels below 10 µSv $yr^{-1}$ based on the derived radioactivity level, an additional layer of soil slope protection of 40 cm or more was needed. Accordingly, the Japanese Ministry of Environment determined the maximum radioactivity level to be 6,000 Bq $kg^{-1}$ for embankments covered with 50 cm uncontaminated soil. Overall, the recycling of decontaminated soil is limited to civil engineering structures in public projects, such as road embankments and coastal levees. Takai et al. (2018) evaluated the associated additional

doses to workers and the public using these structures and demonstrated that additional dose rates would remain below the 1 mSv $yr^{-1}$ threshold corresponding to 6000 Bq $kg^{-1}$.

In Japan, contaminated wastes are disposed under the standard of 8,000 Bq $kg^{-1}$. The volume of decontamination soil having a radioactivity concentration of 8,000 Bq $kg^{-1}$ or below is estimated to be approximately 10 million $m^3$, which corresponds to half of the total amount of decontamination soil generated. The radioactivity concentration of 8,000 Bq $kg^{-1}$ will decrease to 6,000 Bq $kg^{-1}$ in 5 years. Therefore, more than half of the total decontamination soil should become recyclable in at least 5 years. Through the use of pre-treatment activities, such as classification processing, even more decontaminated soil may become recyclable in the non-too distant future (Takai et al., 2018).

*Soil remediation*

Remediation of contaminated soil based on a hot acid treatment was tested for the two most common soil groups found in Fukushima (Parajuli et al., 2016b): Cambisols (i.e. brown forest soils) and Andisols (i.e. soils developed on volcanic ash). Although this method was shown to be effective for the former soil type, this was not the case for the latter. In particular, lime must be added to treat Andisols which must be mixed with untreated and uncontaminated soil prior to being reused for cultivation. Furthermore, to avoid the transfer of residual radiocesium to plants, additives such as zeolite or Prussian blue adsorbents need to be incorporated into the Andisols. The problem associated with this strategy is that, through their ageing, zeolites may increase $^{137}$Cs exchangeability with potassium and accelerate $^{137}$Cs transfer to the cultivated plants over longer time periods (Yamaguchi et al., 2019). These restrictions illustrate the difficulty of finding alternatives to the storage of decontamination soil waste in interim facilities.

## 7. Perspectives for future research

The total estimated decontamination cost would exceed 16 trillion JPY (~128 billion EUR) if all forested areas exposed to radiation dose rates exceeding 1 mSv $yr^{-1}$ were decontaminated. However, decontaminating all of the forested areas would not result in a major ED reduction for the average inhabitant (Yasutaka and Naito, 2016). As almost 70% of the surface area of Fukushima Prefecture is covered with forests (Hashimoto et al., 2012) and forestry is a significant economic activity in the region, future research should prioritize investigating radiocesium dynamics in these regions. In particular, the biological cycling of $^{137}$Cs in forests has now been affected by the decomposition of litter where radiocesium was concentrated shortly after the FDNPP accident (Koarashi et al., 2012). Furthermore, the local population in rural areas of the Fukushima Prefecture enjoy 'satoyama', or the collection of vegetation, including mushrooms, edible wild plants, and firewood from forested landscapes (Prand-Stritzko and Steinhauser, 2018;Nihei, 2016). In addition, approximately 1800 workers are employed by the forest industry in the region (Yasutaka and Naito, 2016). For many of the local inhabitants, the forest, the satoyama, and its harvest are inseparable from their daily lives.

Forest sources were also shown to deliver a significant proportion of contaminated material to the river systems draining the fallout-impact region. The analysis of deposited particulate matter collected in three fallout-contaminated coastal catchments between November 2012 and November 2014 demonstrated that forest sources supplied a mean of 17 % (standard deviation, SD, 10 %) of the sediment transiting these river systems (Laceby et al., 2016). Huon et al. (2018) obtained similar results through the analysis of sediment cores collected between November 2014 and April 2015 in a dam reservoir draining fallout-impacted cultivated and forested landscapes, with the latter supplying a mean of 27% (SD, 6%) of the material deposited in the lake. These conclusions were validated through an analysis of a larger number of sediment samples (n=400) collected in coastal river systems in the

Fukushima region over a longer time period (from November 2011 to November 2017) where a mean of 24% (SD, 21%) of the material transiting these systems was modelled to be derived from forested landscapes (Evrard et al., 2019). Cumulatively, these results demonstrate that forested landscapes represent a potential long-term source of particulate contaminated matter that likely will require diligent management for the foreseeable future.

In cultivated landscapes where the remediation activities were concentrated, the main question is whether or not to restart agricultural production. The removal of the topsoil layer concentrating the radiocesium, the replacement of this material with crushed granite extracted from local quarries and the final mixing of the entire profile to prepare the soils for re-cultivation raises several important questions. For example, to what extent will the residual radiocesium in the soil be transferrable to the plants cultivated on these soils? How will the crushed granite, which was homogenized into the soil, affect the soils fertility? Recent research showed that potassium fertilization is required to maintain productivity when restarting cultivation after decontamination (Kurokawa et al., 2019). Indeed, as it was demonstrated in the current literature review, the reopening of the region after the completion of remediation activities represents a unique situation in history, coupled with unprecedented challenges that require further ongoing investigations.

Although previous dosimetric studies demonstrated that currently the internal exposure of both the local population and the decontamination workers remains minimal, both internal and external exposures of these groups should be studied over longer temporal periods to help understand long-term impacts of this and potentially other nuclear accidents on exposed population groups. More research is also required to understand the fate and dynamics of other longer-lived radionuclides in the Fukushima region including radiocarbon (Paterne et al., 2018;Povinec et al., 2016;Xu et al., 2016), plutonium and uranium isotopes (Jaegler et al., 2018;Zheng et al., 2013;Steinhauser, 2014) as they may be persistent in the environment even though many were emitted only at the trace and ultra-trace levels.

## Conclusions

The quick and early decision of the Japanese authorities to decontaminate FDNPP fallout impacted landscapes was unprecedented. Decontamination activities were rapidly implemented in agricultural and residential areas covering a surface of ~9000 km². These remediation activities produced ~20 million $m^3$ of soil waste in less than 6 years (2013–2019) with an approximate cost of 3 trillion JPY (~24 billion EUR). The strategy of removing the surface layer of the soil concentrating $^{137}Cs$ was shown to be effective in cultivated land when the strategy was applied at a catchment scale to avoid the supply of mobilized contamination from the headwaters. The main current challenges are associated with the treatment and the transport of this waste to the interim storage facilities for the next ~30 years that are being built near the FDNPP. The re-cultivation of the soils after decontamination also raises several concerns. In particular, more information is required regarding soil fertility after decontamination and the potential transfer of the residual $^{137}Cs$ to the plants cultivated on decontaminated fields.

The risks of internal and external radiation dose exposures of the decontamination workers and the local population to exceed the target of 1mSv $yr^1$ appeared to be low during the early post-accidental phase. However, dosimetric monitoring programmes should be carried out to confirm this result over the longer term, particularly after local population returns to the region. Furthermore, as ~75% of the surface exposed to the highest $^{137}Cs$ fallout levels in the Fukushima Prefecture are covered with forests where decontamination was not implemented, the perennial contribution of radiocesium to the river systems draining these mountainous, forested landscapes exposed to typhoons should be investigated. The behaviour and the dynamics of longer-lived radionuclides such as plutonium isotopes

should also be studied in the future as they may persist in the environment for long timescales even though they were emitted at trace and ultra-trace levels.

**Acknowledgements**

This research was funded by the AMORAD project (ANR-11-RSNR-0002), supported by the French National Research Agency (ANR, Agence Nationale de la Recherche). The support of CNRS (Centre National de la Recherche Scientifique, France) and JSPS (Japan Society for the Promotion of Science) in the framework of the Franco-Japanese collaboration project framework (PRC, CNRS-JSPS) is also gratefully acknowledged.

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

**Figures**

Figure 1. Evolution of the number of studies published on radiocesium and Fukushima (including or not a reference to decontamination) in the literature between 2011–2018, according to the Scopus search engine.

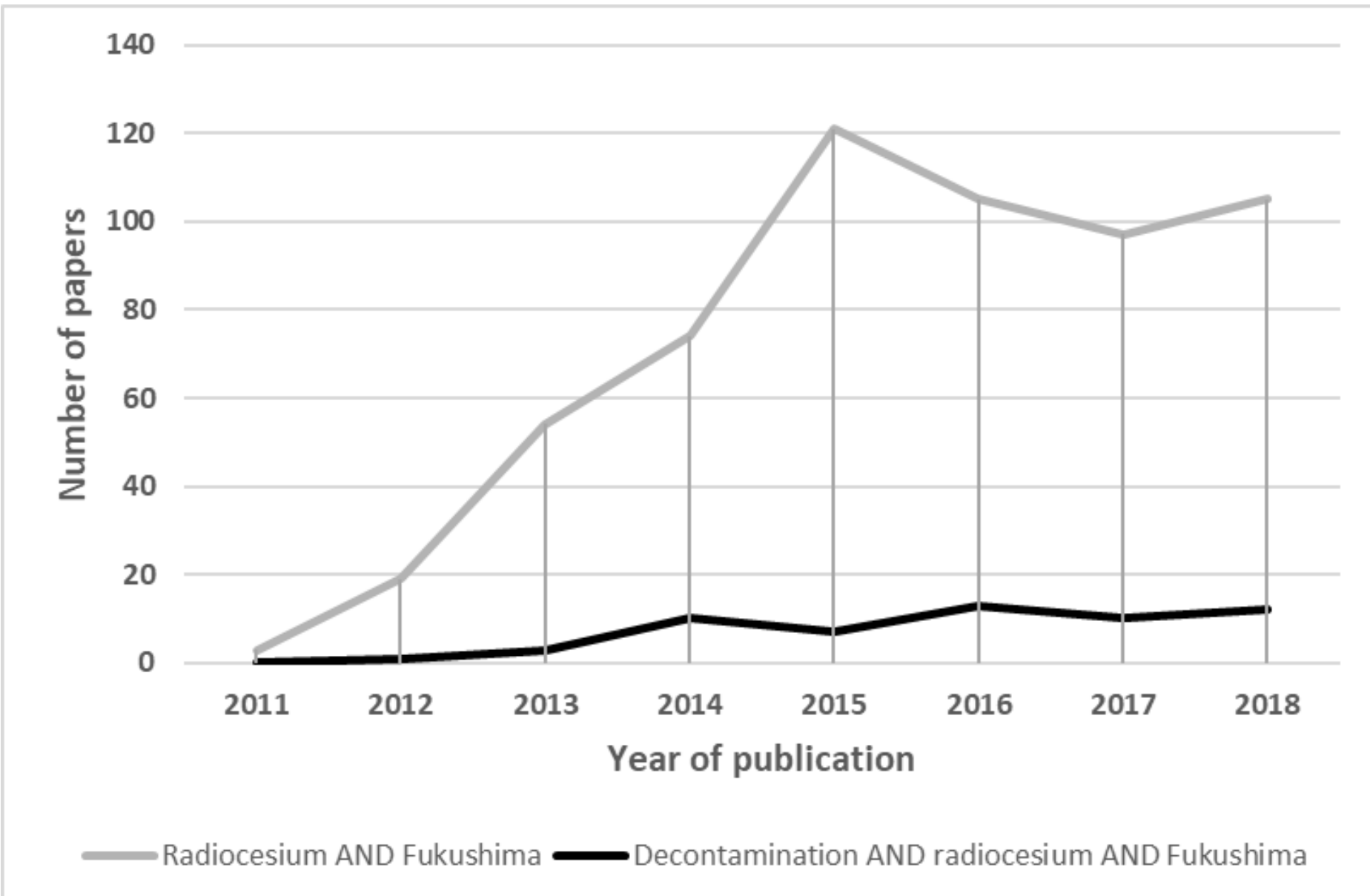

Figure 2. Location of the Fukushima Prefecture in Japan (Inset Map) and the location of the Special Decontamination Zones (SDZs) and the Intensive Contamination Survey Areas (ICAs).

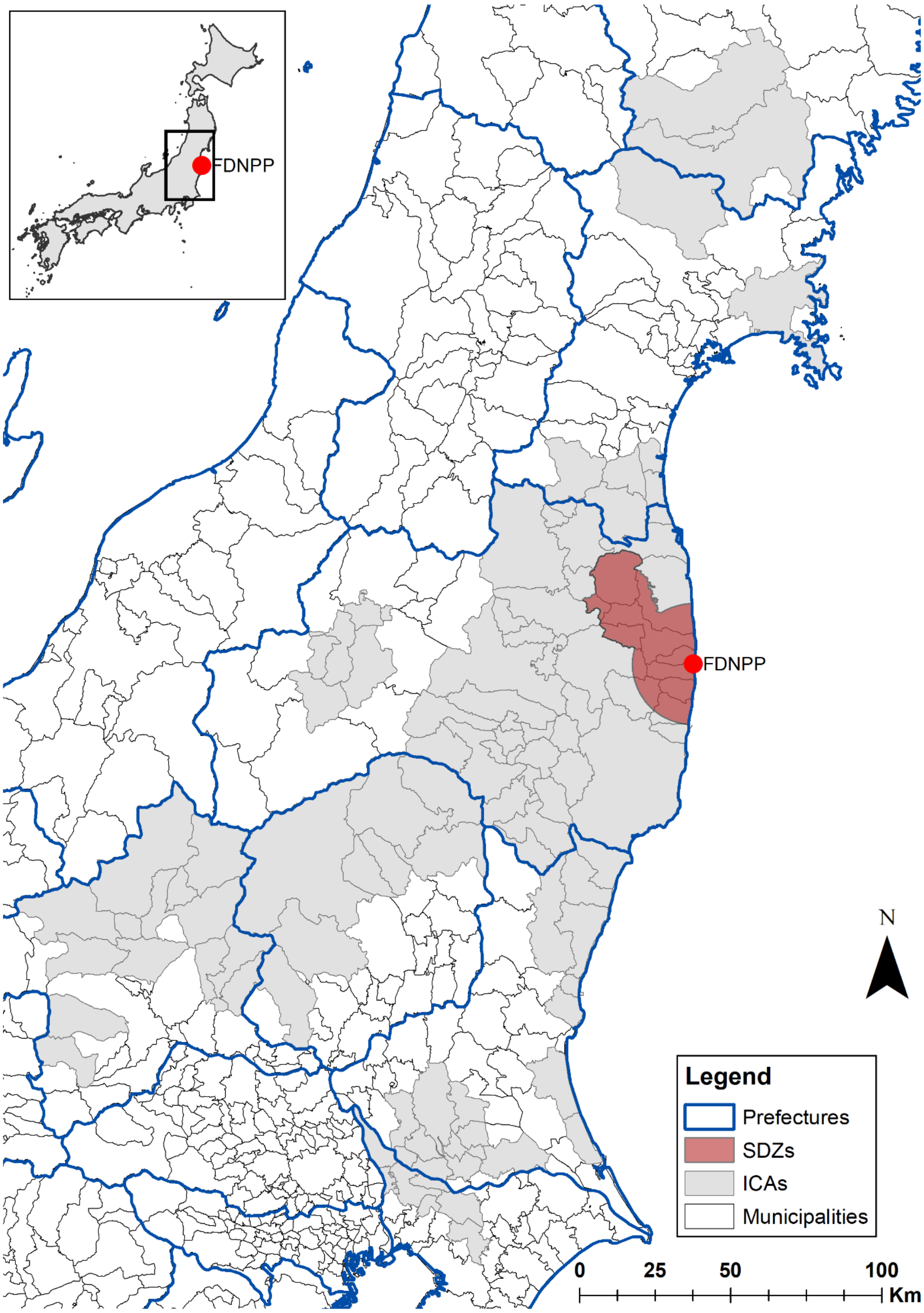

Figure 3. Illustration of the different steps of remediation activities in cultivated land in Fukushima: (a) removal of the 5-cm topsoil layer concentrating most of the radiocesium (November 2013); (b) addition of a crushed granite layer on top of the residual soil profile (May 2014); (c) final mixing of the entire profile to prepare re-cultivation (March 2019). Pictures were taken by the authors in the Iitate Village.

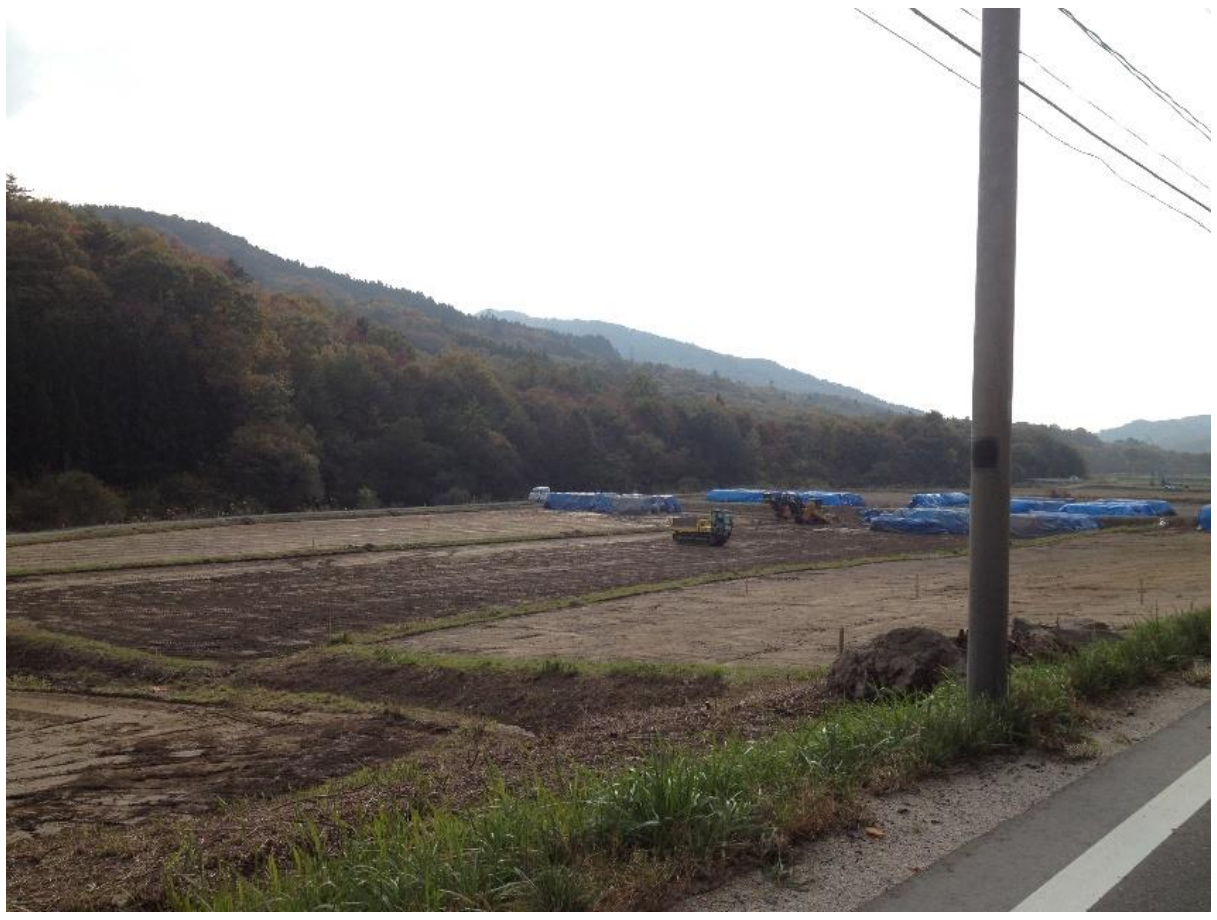
(a)

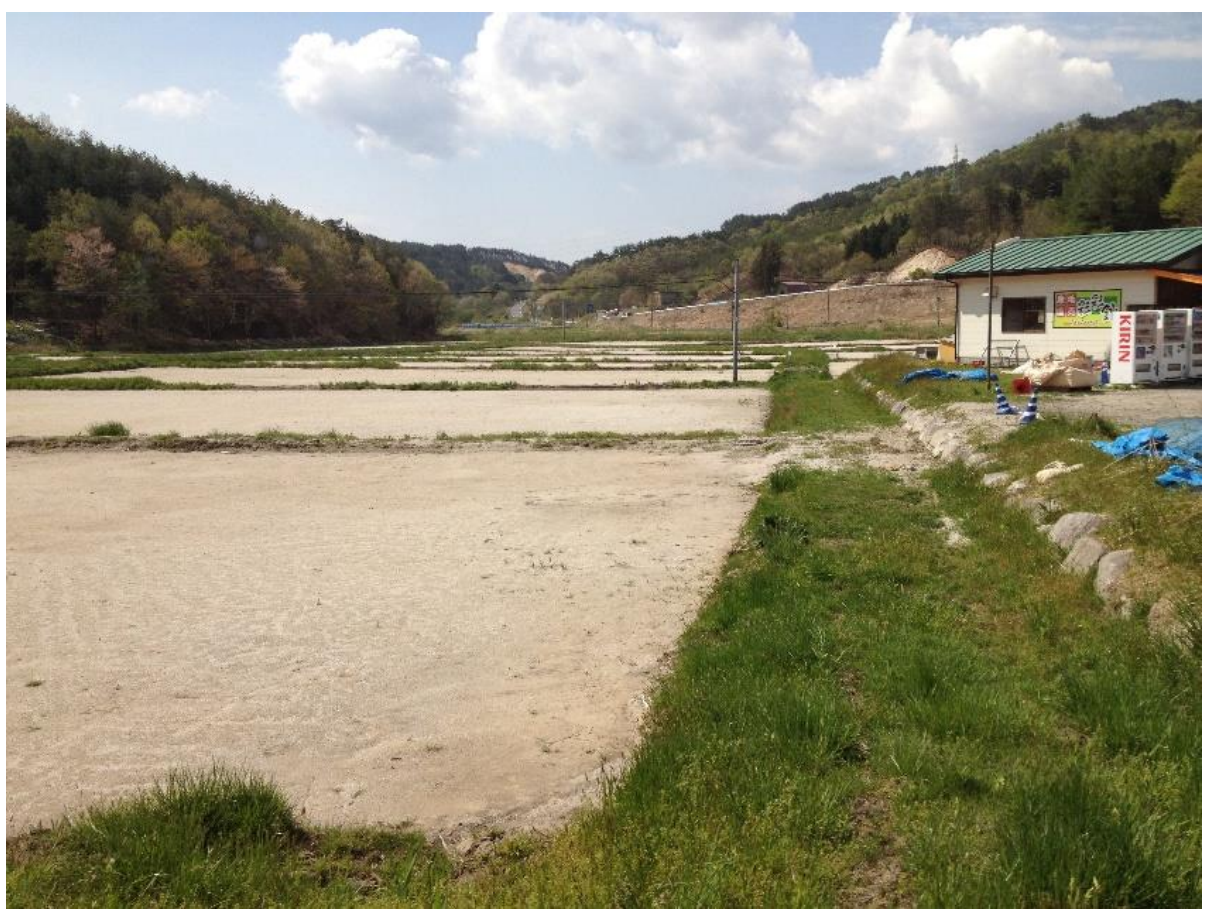
(b)

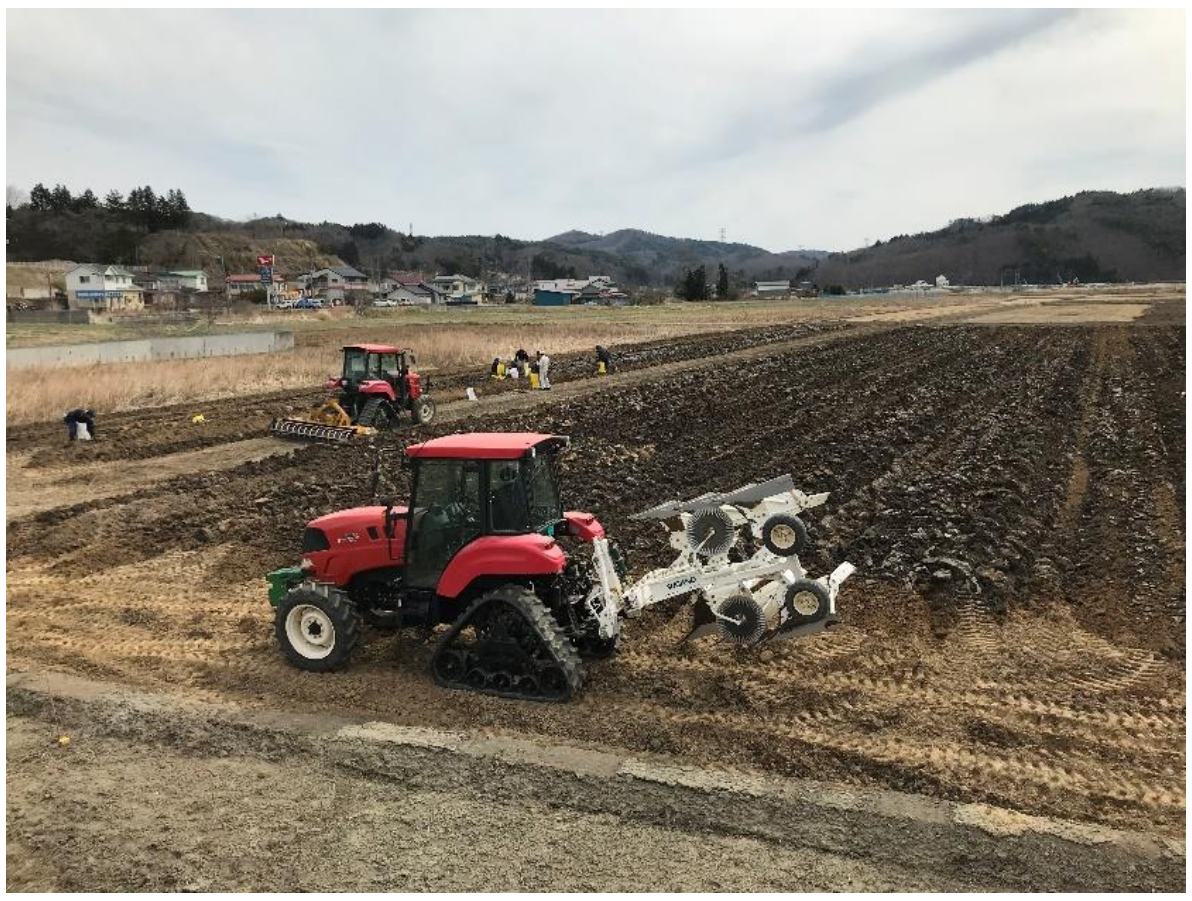
(c)

Figure 4. Illustration of the 20-m buffer zone decontaminated in forested areas in the vicinity of houses. Example from Iitate Village (Sasu district).

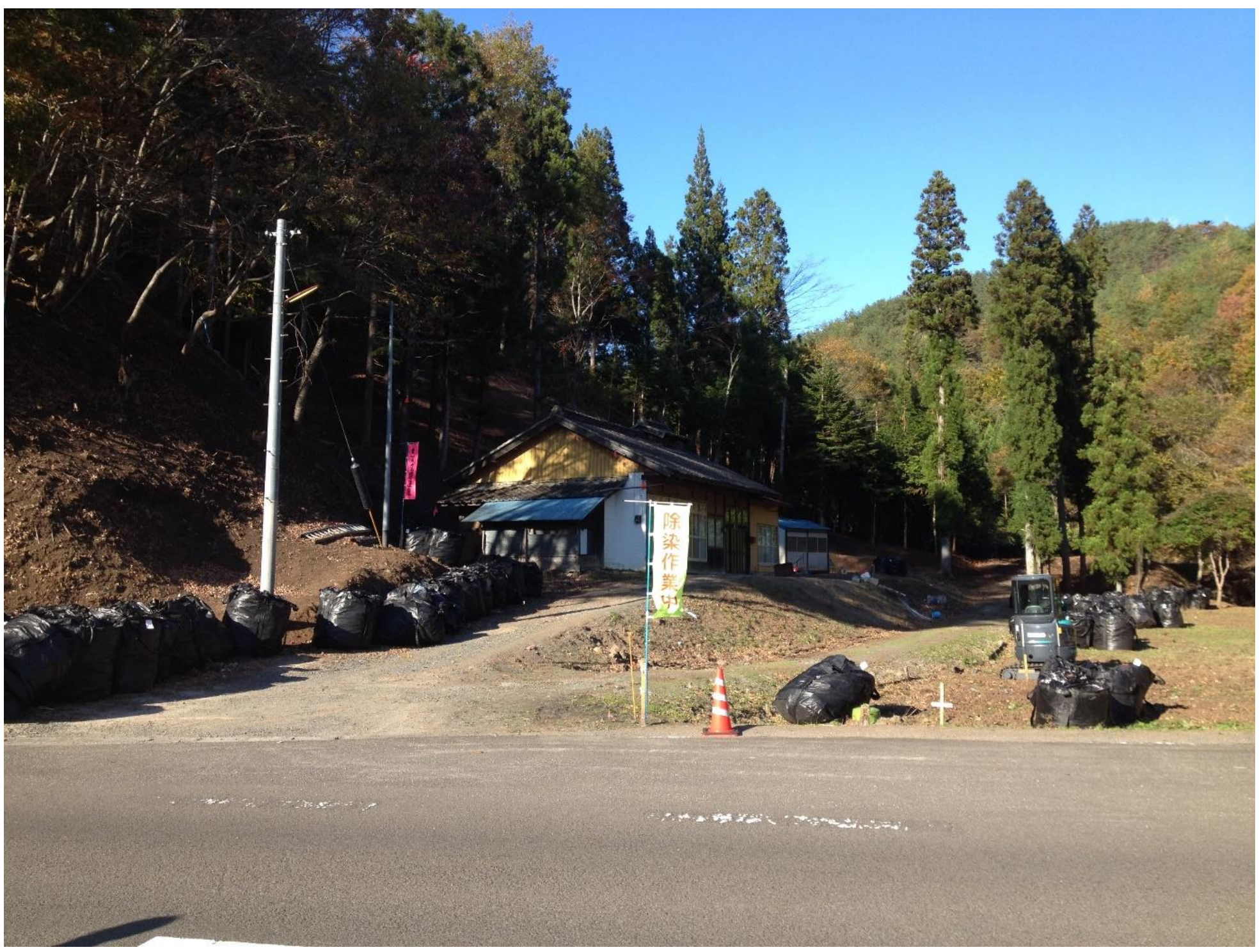

Figure 5. Comparison of annual and individual radioactive dose rates to which the population may be exposed, based on a compilation of data (Commissariat à l'Energie Atomique et aux Energies Alternatives, 2016;Harada et al., 2014).

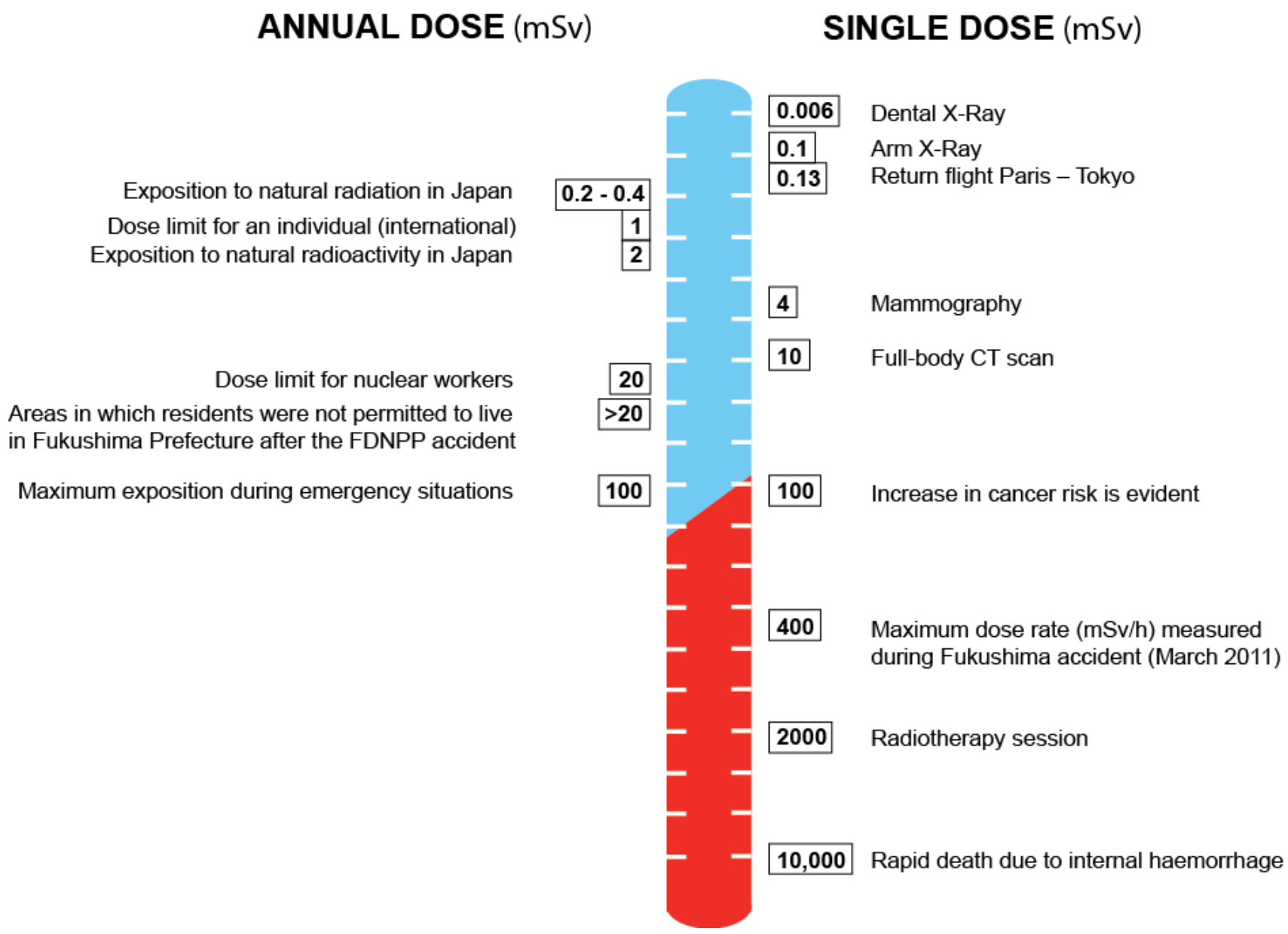

Figure 6. Temporary storage facilities for radioactive waste in Iitate Village, in the Fukushima Prefecture.

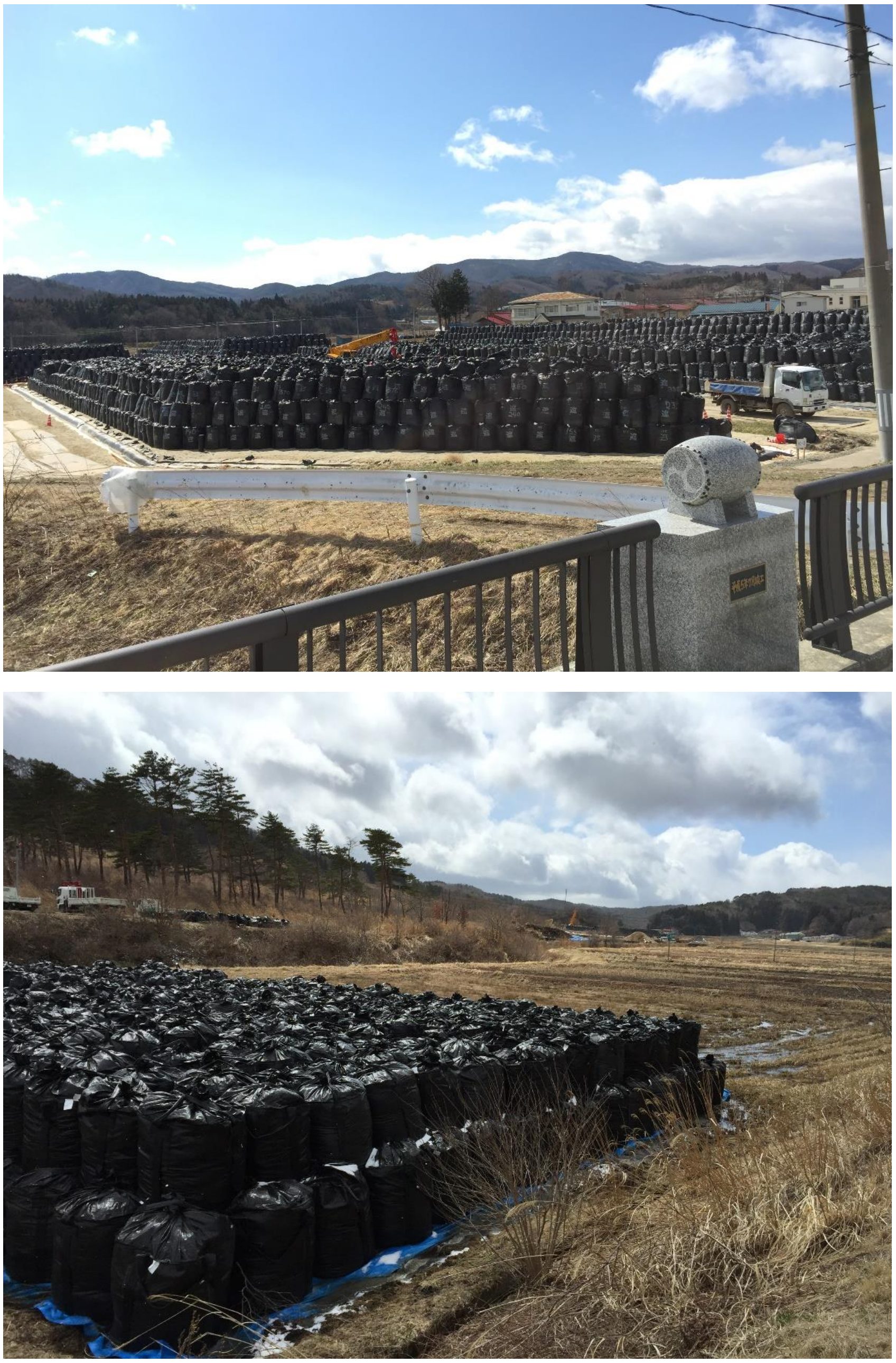

944 Figure 7. Simplified diagram showing the functioning of an incineration plant treating decontamination
945 waste in Fukushima, modified after Parajuli et al. (2013) and Fujiwara et al. (2017).

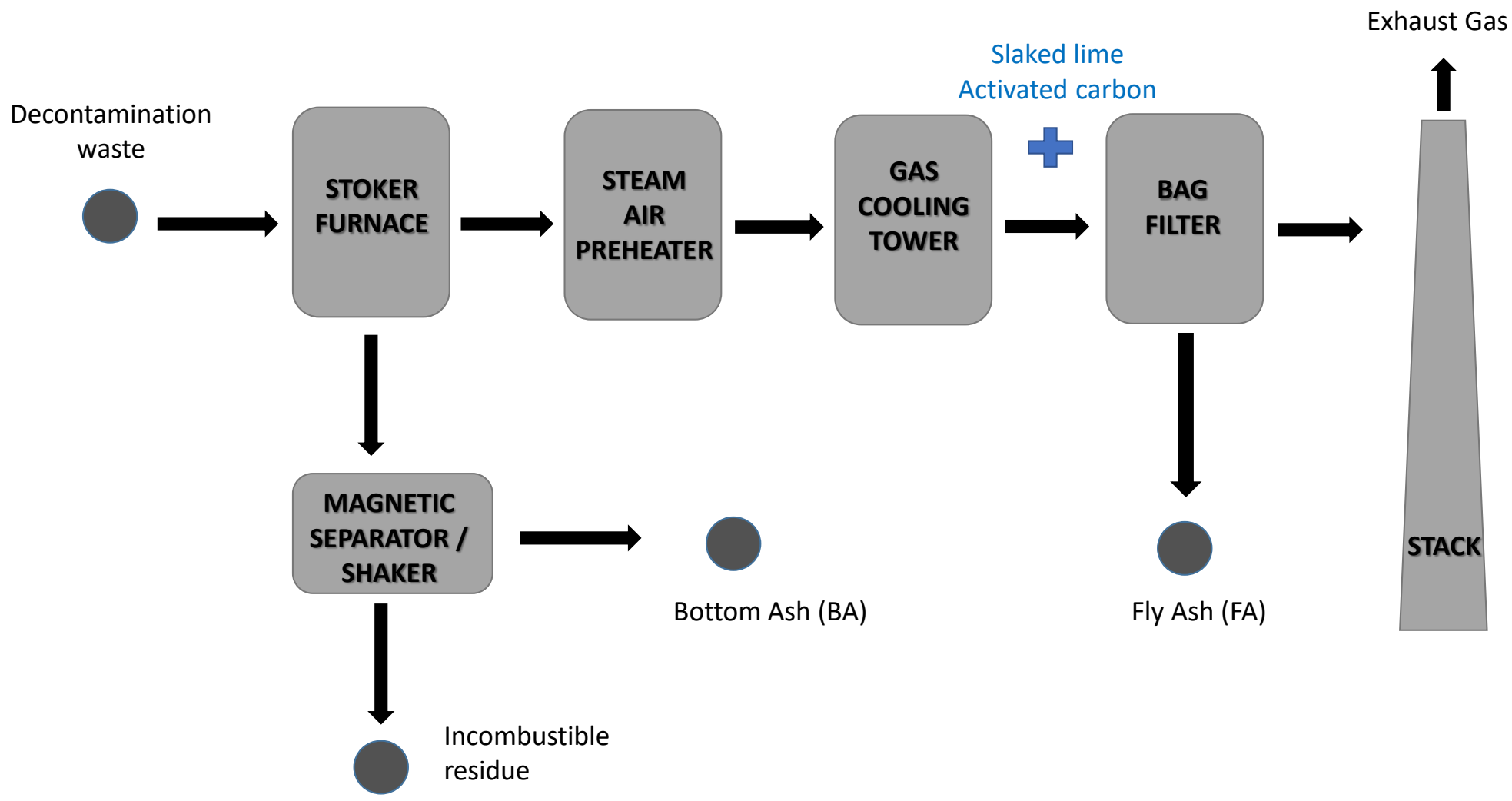

946

## Tables

Table 1. Selection of official websites from the Japanese authorities providing information on the remediation works and their impact. (Last access to these websites on March 8, 2019).

| Authority | Type of information | Webpage |
|---|---|---|
| Fukushima Prefecture | Environmental restoration | https://www.pref.fukushima.lg.jp/site/portal-english/list382.html |
| Ministry of Environment | Environmental Remediation | http://josen.env.go.jp/en/ |
| Ministry of Economy, Trade and Industry | Fukushima Today webpage | http://www.meti.go.jp/english/earthquake/index.html |
| Nuclear Regulation Authority | Monitoring information | https://radioactivity.nsr.go.jp/en/ |

Table 2. Unit costs and effectiveness of decontamination measures implemented in the SDZ and ICAs after Yasutaka and Naito (2016).

| Code | Measure | Effectiveness range | Unit cost 10kJPY/ha (EUR) | Nb containers/ha | Target area |
|---|---|---|---|---|---|
| **Agricultural land** | | | | | |
| A1 | Cut weeds/remove 5cm topsoil/cover soil | 0.34 – 0.80 | 950 (7600) | 815 | SDZ |
| A2 | Cut weeds/remove 5 cm topsoil | 0.34 – 0.80 | 625 (5000) | 815 | SDZ |
| A3 | Interchange topsoil and subsoil/add zeolite – K | 0.34 – 0.80 | 310 (2500) | 0 | SDZ and ICA |
| A4 | Ploughing with zeolite and K | 0.21 – 0.50 | 33 (265) | 0 | SDZ and ICA |
| **Forest** | | | | | |
| F1 | Remove litter and humus | 0.19 – 0.59 | 745 (6000) | 530 | SDZ |
| F2 | Remove litter | 0.10 – 0.30 | 280 (2250) | 260 | ICA |
| **Roads** | | | | | |
| R1 | Shot-blasting, cleaning ditches | 0.15 – 0.66 | 480 (4000) | 30 | SDZ |
| R2 | Cleaning roads and ditches | 0.08 – 0.33 | 240/km (2000) | 88 | ICA |
| **Buildings** | | | | | |
| B1 | Full decontamination | 0.29 – 0.70 | 1750–3500 (14,000–30,000) | 150 | SDZ and ICA |
| B2 | Local decontamination | 0.15 – 0.35 | 125–250 (1000–2000) | 11 | ICA |

Table 3. Unit costs estimated for waste management and storage after Yasutaka and Naito (2016) in the ICAs.

| Measure | Unit cost (JPY) | Unit cost (EUR) |
|---|---|---|
| Storage container | 8000 | 65 |
| Transport from decontamination site to temporary storage site | 3100 / container | 25 / container |
| Temporary storage site | 20,000 / container | 160 /container |
| Transport from temporary storage site to interim storage facility | 3800 – 16,000 / container | 30 – 130 /container |
| *Treatment at interim storage facility* | | |
| Combustible volume reduction | 2000 / container | 16 /container |
| Storage of combustible incineration residue | 100,000 / container | 800 /container |
| Storage of incombustibles | 30,000 / container | 240 /container |