# Peer review of "Effectiveness of landscape decontamination following the Fukushima nuclear accident: A review"

_SOIL, 2019_

## Referee Comment (RC1) · Anonymous Referee #1 · 2 Aug 2019

SOIL-2019-43 Strategies and effectiveness of land decontamination in the region affected by radioactive fallout from the Fukushima nuclear accident: A review

This review describes the effects of the Fukushima Dai-ichi Nuclear Power Plant (FDNPP) accident in March 2011. This review describes the spatial extent of the decontaminated zone and the remediation strategies in different environments (including schedule and costs). The issues of the impacts of remediation activities on dosimetry and large volume of waste generated are also discussed. They conclude that the strategy of removing the surface layer of the soil concentrating 137Cs was effective

in cultivated land at a catchment scale avoiding its transformation into source of contamination. Discussion and conclusion they are extremely interesting and provocative. This manuscript adheres to the journal's standards. The research meets the applicable standards for the research integrity. The research output, in terms of novelty, scores good uniqueness. The level of clarity is well above the threshold of acceptability. Potentially, its potential impact upon the international scientific community of reference is good. The article is presented in an intelligible manner. This work is interesting and deserves to be published.

Title: 20 words. It can be shortened Abstract: OK. Keywords: REVISE. The keywords, together with title and abstract function in a system comparable to a chain reaction. Once the keywords have assisted the Reader find the suitable paper and its title has fruitfully drawn in the attention, it is up to the abstract to further activate the interest and keep their curiosity. So, these three elements must work together and not replicate each other. Introduction: OK Conclusion: OK. If confirmed. Figures: REVISE. Figure 1 unnecessary (in case, move to Supplementary Material), I would suggest to shift Figure 6 here. Figure 2 is quite useless (please, add geographical coordinates). This figure is rather unnecessary, a kmz file would be more useful. References: REVISE.

In particular (page.row): 2.63 This section, including Figure 1, it is purely of a methodological nature, being a review, and, in my opinion, it takes away the bite from the paper. 2.72 Also in my opinion, a review is the right interface where 'grey literature' becomes scientific literature, sensu stricto. Precisely, it must be the responsibility of the authors to insert unpublished and reliable information. The fact that a job has been peer-reviewed, per se, is not a guarantee of total reliability. As, on the other hand, official documents (published in languages other than English, as mostly in Japanese in this case) are certainly not unreliable. My advice is to insert that grey literature useful to enrich the review of data and interpretations. Converting Table 1 into individual references. 3.96 I would begin by telling, briefly, what happened. Mentioning, for instance, the International Nuclear and Radiological Event

Scale (INES) introduced by the International Atomic Energy Agency. And, discussing, comparatively (e.g. Ivanov et al. 1997, Rosén et al. 1999. And the, already quoted, doi>10.1016/j.scitotenv.2013.10.029), the two Level 7 INES Major accidents. 3.113 I would mention that the sievert is a derived unit of ionizing radiation dose in the International System of Units and is a measure of the human health effect of ionizing radiation 4.144 what interchanged means, exactly? 4.163 A term of comparison would help the Reader. For instance, the whole EU budget was at some €37 billion in 2017. 5.192 Please, discuss this issue comparatively (e.g. doi> 10.1021/es980788+) 6.258 Please, discuss this issue comparatively (e.g. Santschl et al. 1990) 12.505 This section deserve more room and comparative discussion (e.g. Rosén et al. 1999)

References Ivanov YA et al. 1997. Migration of 137Cs and 90Sr from Chernobyl fallout in Ukrainian, Belarussian and Russian soils. J. Environ. Radioactivity 35, I-21 Rosén K et al. 1999. Migration of radiocaesium in Swedish soil profiles after the Chernobyl accident, 1987-1995. J. Environ. Radioactivity 46, 45-66 Santschl PH et al. 1990. The self-cleaning capacity of surface waters after radioactive fallout evidence from European waters after Chernobyl, 1986-1988. Environ. Sci. Techn. 24, 519-527

The manuscript represent a substantial contribution to scientific progress within the scope of SOIL (interdisciplinary, mainly). The results are discussed in a thorough and balanced way (consideration of related and relevant work, including appropriate references need some reworking). Scientific results and conclusions are presented in a clear, concise, and quite well-structured way.

Recommendation: Major revision is required

---

## Short Comment (SC1) · 28 Aug 2019

[revised manuscript text omitted]

---

## Referee Comment (RC2) · Anonymous Referee #2 · 1 Oct 2019

The review paper entitle " Strategies and effectiveness of land decontamination in the region affected 1 by radioactive fallout 2 from the Fukushima nuclear accident: A review"

It is a very well written review, very informative with plenty of details. A large number of publications are given.

With this review, it is becoming obvious how the remediated actions taken from Japanese authorities had crucially resulted in low effective doses to the population

Furthermore, the remediated actions taken in Japan will remain a great lesson in case of a similar accident.

All aspects or remediation actions in various environments are covered as well as their effectiveness. Furthermore, perspective of future research is given too.

The manuscript can be published as it is.

---

## Author Comment (AC1) · 2 Oct 2019

Many thanks for providing a detailed review of the manuscript and for your general encouraging comments. We will definitely take into account all your detailed suggestions when revising the manuscript. In particular, we will consider the addition of a kmz file with the location data and the inclusion of additional grey literature data.

---

## Author Comment (AC2) · 2 Oct 2019

Many thanks for your detailed feedback on the text and for sharing your expertise on the topic. We will definitely take all your suggestions into account when preparing a revised version of the manuscript.

---

## Author Comment (AC3) · 2 Oct 2019

Many thanks for sharing your positive evaluation of this work. We are glad that you appreciated it. We will nevertheless take the comments provided by the reviewer #1 and Dr. Yves Thiry into account to prepare a revised version of this manuscript.

---

## Referee Comment (RC3) · Anonymous Referee #3 · 20 Oct 2019

Manuscript "Strategies and effectiveness of land decontamination in the region affected by radioactive fallout from the Fukushima nuclear accident: A review", by Evrard et alii is a very interesting report of the nuclear accident occurred in Japan eight years ago and, in particular, of the countermeasures adopted to safeguard the well-being of the populations involved. I liked reading it, also because it is well written and contains a lot of ancillary information to the main subject however useful to understand the enormity of the problem. I suggest only the following few additions, which I think can further improve the quality of work: - without absolutely making a comparison between the

two events, at least in terms of soil contamination levels could some Chernobyl data be cited? - Lines 507-509: please provide an explanation of why hot acid treatment does not work with Andisols, and why these soils need lime to be reused for cultivation. - Lines 526-528: can you say something about the restrictions to which people in the contaminated areas has had to adhere in terms of collection of mushrooms and other products of the understory or use of firewood?

––––––––––––––––––––––––––––––

---

## Author Comment (AC4) · 20 Oct 2019

Many thanks for these positive comments and for these suggested minor edits. We will definitely take them all into account when preparing a revised version of the manuscript.

---

## Author Response (AR1)

| Reviewer's comment | Authors' reply |
|---|---|
| **Anonymous reviewer # 1** | |
| SOIL-2019-43 Strategies and effectiveness of land decontamination in the region affected by radioactive fallout from the Fukushima nuclear accident: A review

This review describes the effects of the Fukushima Dai-ichi Nuclear Power Plant (FDNPP) accident in March 2011. This review describes the spatial extent of the decontaminated zone and the remediation strategies in different environments (including schedule and costs). The issues of the impacts of remediation activities on dosimetry and large volume of waste generated are also discussed. They conclude that the strategy of removing the surface layer of the soil concentrating 137Cs was effective in cultivated land at a catchment scale avoiding its transformation into source of contamination.

Discussion and conclusion they are extremely interesting and provocative.

This manuscript adheres to the journal's standards. The research meets the applicable standards for the research integrity. The research output, in terms of novelty, scores good uniqueness. The level of clarity is well above the threshold of acceptability. Potentially, its potential impact upon the international scientific community of reference is good. The article is presented in an intelligible manner. This work is interesting and deserves to be published. | Many thanks for these encouraging general comments! |
| Title: 20 words. It can be shortened | As recommended, we have shortened the title of the manuscript (P.1 L.3). |
| Abstract: OK. | OK |
| Keywords: REVISE. The keywords, together with title and abstract function in a system comparable to a chain reaction. Once the keywords have assisted the Reader find the suitable paper and its title has fruitfully drawn in the attention, it is up to the abstract to further activate the interest and keep their curiosity. So, these three elements must work together and not replicate each other. | Thank you for this comment - the 'nuclear accident' keyword has been removed as it duplicated the information contained in the title. It was replaced with 'phytoextraction' (P. 1 L. 43). |
| Introduction: OK | OK |
| Conclusion: OK. If confirmed. | OK |
| Figures: REVISE. Figure 1 unnecessary (in case, move to Supplementary Material), I would suggest to shift Figure 6 here. | We somewhat disagree with the reviewer, and we think that providing Fig. 1 is informative for the readership. Furthermore, the other |

| | |
|---|---|
| Figure 2 is quite useless (please, add geographical coordinates). This figure is rather unnecessary, a kmz file would be more useful. | reviewers/commenters did not request to remove Fig. 1. We also do think that Fig. 2 is important. Accordingly, we have updated figure 2 to now include geographic coordinates. In addition, we now also provide a kmz file as Supplementary Material. Many thanks for this suggestion. |
| References: REVISE. | |
| In particular (page.row): 2.63 This section, including Figure 1, it is purely of a methodological nature, being a review, and, in my opinion, it takes away the bite from the paper. | We agree with the reviewer that this is not the main message of the paper, although we do think that this section sets the scene and puts this research in context. Furthermore, this approach of grounded a review in the literature is quite common in review papers. Of note, the other reviewers did not request to change or remove this paragraph. |
| 2.72 Also in my opinion, a review is the right interface where 'grey literature' becomes scientific literature, sensu stricto. Precisely, it must be the responsibility of the authors to insert unpublished and reliable information. The fact that a job has been peer-reviewed, per se, is not a guarantee of total reliability. As, on the other hand, official documents (published in languages other than English, as mostly in Japanese in this case) are certainly not unreliable. My advice is to insert that grey literature useful to enrich the review of data and interpretations. Converting Table 1 into individual references. | Of note, we integrated in the text the most up-to-date information from these grey sources that is of interest to the readership of our manuscript (more than 10 references to grey literature documents were made in this manuscript). Most information available in the grey literature sources referred to in Table 1 consist of guidelines provided by the Japanese authorities. |
| 3.96 I would begin by telling, briefly, what happened. Mentioning, for instance, the International Nuclear and Radiological Event Scale (INES) introduced by the International Atomic Energy Agency. And, discussing, comparatively (e.g. Ivanov et al. 1997, Rosén et al. 1999. And the, already quoted, doi>10.1016/j.scitotenv.2013.10.029), the two Level 7 INES Major accidents. | We added a short description of the events that occurred at FDNPP in 2011 (P.3, LL.109-116).

We agree that the INES level associated with the accident is very important to mention. This was added to the very beginning of the manuscript (P.1 LL.48-49).

As the objective of the current manuscript is not to compare the Chernobyl and the Fukushima accidents, we added this information in the abstract with relevant references for those readers who would be interested in this topic (PP.2-3; LL.89-95). |
| 3.113 I would mention that the sievert is a derived unit of ionizing radiation dose in the International System of Units and is a measure of the human health effect of ionizing radiation | Done, on P.3 LL. 131-132. |

| | |
|---|---|
| 4.144 what interchanged means, exactly? | Replaced (modified on P. 4 L. 163). |
| 4.163 A term of comparison would help the Reader. For instance, the whole EU budget was at some C37 billion in 2017 | A comparison with the EU budget in 2017 was added (P.5, LL.186-188). |
| 5.192 Please, discuss this issue comparatively (e.g. doi> 10.1021/es980788+) | Again, the main goal was to review the current strategies and effectiveness of decontamination efforts after the Fukushima accident. Adding in a comprehensive comparison to Chernobyl is beyond the scope of this review. To address these and other comments, we did note how the fallout situations are different in these two regions which would complicate a direct comparison of the remediation and decontamination efforts (PP.2-3; LL.89-95). |
| 6.258 Please, discuss this issue comparatively (e.g. Santschl et al. 1990) | Please see our other comments regarding the comparison with the Chernobyl situation. |
| 12.505 This section deserve more room and comparative discussion (e.g. Rosén et al. 1999) | Please see our other comments regarding the comparison with the Chernobyl situation. |
| References Ivanov YA et al. 1997. Migration of 137Cs and 90Sr from Chernobyl fallout in Ukrainian, Belarussian and Russian soils. J. Environ. Radioactivity 35, I-21 Rosén K et al. 1999. Migration of radiocaesium in Swedish soil profiles after the Chernobyl accident, 1987-1995. J. Environ. Radioactivity 46, 45-66 Santschl PH et al. 1990. The self-cleaning capacity of surface waters after radioactive fallout evidence from European waters after Chernobyl, 1986-1988. Environ. Sci. Techn. 24, 519-527 | Again, although we acknowledge that these articles are of a wide interest, the objective of this review was to focus on the lessons learnt from the Fukushima case study, which is very different on many aspects from the Chernobyl experience. Although we agree that a comparison of the strategies of remediation and their effectiveness in Fukushima vs. in Chernobyl would have a wide interest, our opinion is that it falls beyond the scope of the current review. However, these references were added to the current manuscript (PP.2-3; LL.89-95). |
| The manuscript represent a substantial contribution to scientific progress within the scope of SOIL (interdisciplinary, mainly). The results are discussed in a thorough and balanced way (consideration of related and relevant work, including appropriate references need some reworking). Scientific results and conclusions are presented in a clear, concise, and quite well-structured way. Recommendation: Major revision is required | Many thanks for these encouraging general comments! |
| **Yves Thiry's comments (annotated manuscript)** | |
| Nice review at the good level for an educational communication and popularization on the issue and the challenging questions that remain. I included directly in the PDF file some suggestions of additional precisions or questions that would merit clarifications. Please also note the supplement to this comment:https://www.soil-discuss.net/soil-2019-43/soil-2019-43-SC1-supplement.pdf | Many thanks for these positive comments and for sharing these suggestions in the pdf file. All the points were addressed in the text (except the comparison with the Chernobyl situation, please see our reply to Reviewer 1 on this topic). |

| **Anonymous reviewer # 2** | |
|---|---|
| The review paper entitle "Strategies and effectiveness of land decontamination in the region affected 1 by radioactive fallout 2 from the Fukushima nuclear accident: A review"

It is a very well written review, very informative with plenty of details. A large number of publications are given.

With this review, it is becoming obvious how the remediated actions taken from Japanese authorities had crucially resulted in low effective doses to the population

Furthermore, the remediated actions taken in Japan will remain a great lesson in case of a similar accident.

All aspects or remediation actions in various environments are covered as well as their effectiveness. Furthermore, perspective of future research is given too.

The manuscript can be published as it is. | Many thanks for this positive review! |
| **Anonymous reviewer # 3** | |
| Manuscript "Strategies and effectiveness of land decontamination in the region affected by radioactive fallout from the Fukushima nuclear accident: A review", by Evrard et al is a very interesting report of the nuclear accident occurred in Japan eight years ago and, in particular, of the countermeasures adopted to safeguard the well-being of the populations involved. I liked reading it, also because it is well written and contains a lot of ancillary information to the main subject however useful to understand the enormity of the problem. | Many thanks for these positive comments. |
| I suggest only the following few additions, which I think can further improve the quality of work: - without absolutely making a comparison between the two events, at least in terms of soil contamination levels could some Chernobyl data be cited? | Added (P.2, LL. 55-58). |
| - Lines 507-509: please provide an explanation of why hot acid treatment does not work with Andisols, and why these soils need lime to be reused for cultivation.- | Lime is needed to readjust the pH to reasonable levels after acid treatment for allowing recultivation on these soils, Andisols were more sensitive to this treatment than Cambisols (P.12, LL.539-540). |
| Lines 526-528: can you say something about the restrictions to which people in the contaminated areas has had to adhere in terms of collection of mushrooms and other products of the understory or use of firewood? | Added (P.13; LL.559-563). |

[revised manuscript text omitted]